# *TERT* Expression in Wilms Tumor Is Regulated by Promoter Mutation or Hypermethylation, WT1, and N-MYC

**DOI:** 10.3390/cancers14071655

**Published:** 2022-03-25

**Authors:** Carolyn M. Jablonowski, Hyea Jin Gil, Emilia M. Pinto, Prahalathan Pichavaram, Andrew M. Fleming, Michael R. Clay, Dongli Hu, Christopher L. Morton, Shondra M. Pruett-Miller, Baranda S. Hansen, Xiang Chen, Karissa M. Dieseldorff Jones, Yanling Liu, Xiaotu Ma, Jun Yang, Andrew M. Davidoff, Gerard P. Zambetti, Andrew J. Murphy

**Affiliations:** 1Department of Surgery, St. Jude Children’s Research Hospital, 262 Danny Thomas Place, Mail Stop 133, Memphis, TN 38105, USA; carolyn.jablonowski@stjude.org (C.M.J.); hyeajingil@yahoo.com (H.J.G.); prahalathan.pichavaram@stjude.org (P.P.); andrew.fleming@stjude.org (A.M.F.); dongli.hu@stjude.org (D.H.); chris.morton@stjude.org (C.L.M.); jun.yang2@stjude.org (J.Y.); andrew.davidoff@stjude.org (A.M.D.); 2Department of Pathology, St. Jude Children’s Research Hospital, Memphis, TN 38105, USA; emilia.pinto@stjude.org (E.M.P.); gerard.zambetti@stjude.org (G.P.Z.); 3Department of Pathology, University of Colorado Anschutz, Aurora, CO 80045, USA; michael.clay@cuanschutz.edu; 4Department of Cell and Molecular Biology, St. Jude Children’s Research Hospital, Memphis, TN 38105, USA; shondra.miller@stjude.org (S.M.P.-M.); baranda.hansen@stjude.org (B.S.H.); 5Department of Computational Biology, St. Jude Children’s Research Hospital, Memphis, TN 38105, USA; xiang.chen@stjude.org (X.C.); karissa.dieseldorffjones@stjude.org (K.M.D.J.); yanling.liu@stjude.org (Y.L.); xiaotu.ma@stjude.org (X.M.); 6Division of Pediatric Surgery, Department of Surgery, University of Tennessee Health Science Center, Memphis, TN 38105, USA

**Keywords:** Wilms tumor, *WT1*, *TERT*, telomerase, *MYCN*, N-MYC, anaplasia, blastema, kidney

## Abstract

**Simple Summary:**

The telomerase enzyme adds repetitive genetic sequences to the ends of chromosomes called telomeres to prevent cellular senescence. Gain of telomerase function is one of the hallmarks of human cancer. The telomerase protein is coded by the gene *TERT* and increased *TERT* RNA levels have been associated with disease relapse in Wilms tumor, the most common kidney cancer of childhood. This study aimed to determine the mechanisms of increased *TERT* expression in Wilms tumor. This study found mutations in the *TERT* promoter, increased methylation of the *TERT* promoter, and genomic copy number amplifications of *TERT* as potential mechanisms of TERT activation. Conversely, this study found that inactivating *WT1* mutation was associated with low *TERT* RNA levels and telomerase activity. N-MYC overexpression in Wilms tumor cells resulted in increased *TERT* promoter activity and *TERT* transcription. *TERT* transcription is associated with molecular and histologic subgroups in Wilms tumor and telomere-targeted therapies warrant future investigation.

**Abstract:**

Increased *TERT* mRNA is associated with disease relapse in favorable histology Wilms tumor (WT). This study sought to understand the mechanism of increased *TERT* expression by determining the association between *TERT* and WT1 and N-MYC, two proteins important in Wilms tumor pathogenesis that have been shown to regulate *TERT* expression. Three out of 45 (6.7%) WTs and the corresponding patient-derived xenografts harbored canonical gain-of-function mutations in the *TERT* promoter. This study identified near ubiquitous hypermethylation of the *TERT* promoter region in WT compared to normal kidney. WTs with biallelic inactivating mutations in *WT1* (7/45, 15.6%) were found to have lower *TERT* expression by RNA-seq and qRT-PCR and lower telomerase activity determined by the telomerase repeat amplification protocol. Anaplastic histology and increased percentage of blastema were positively correlated with higher *TERT* expression and telomerase activity. In vitro shRNA knockdown of *WT1* resulted in decreased expression of *TERT*, reduced colony formation, and decreased proliferation of WiT49, an anaplastic WT cell line with wild-type *WT1*. CRISPR-Cas9-mediated knockout of *WT1* resulted in decreased expression of telomere-related gene pathways. However, an inducible *Wt1*-knockout mouse model showed no relationship between *Wt1* knockout and *Tert* expression in normal murine nephrogenesis, suggesting that WT1 and TERT are coupled in transformed cells but not in normal kidney tissues. N-MYC overexpression resulted in increased *TERT* promoter activity and *TERT* transcription. Thus, multiple mechanisms of *TERT* activation are involved in WT and are associated with anaplastic histology and increased blastema. This study is novel because it identifies potential mechanisms of *TERT* activation in Wilms tumor that could be of therapeutic interests.

## 1. Introduction

Wilms tumor (WT) is the most common childhood kidney cancer and represents 6–7% of pediatric cancer cases [1]. Although overall outcomes for WT are excellent, patients with diffuse anaplasia (unfavorable histology) or patients with favorable histology who experience disease relapse continue to have poor overall survival [1]. In International Society of Pediatric Oncology (SIOP) protocols, patients with blastemal predominance after neoadjuvant chemotherapy have inferior outcomes, warranting intensification of adjuvant chemotherapy [2]. Focused research efforts are needed to improve outcomes for these specific patient populations. The finding of diffuse anaplasia is attributed to acquired development and expansion of *TP53* mutant clones within an initially favorable histology WT [3,4,5]. Additionally, *MYCN* amplification and mutation (gain-of-function P44L hotspot mutation) are suspected to contribute to the pathogenesis of anaplastic WT and are among the most common genetic alterations found in Wilms tumor [6,7]. Approximately 50% of patients with favorable histology who experience disease relapse die from their disease [8,9]. Adverse biological features of Wilms tumor associated with relapse include mutations in *TP53*, *SIX1* or *SIX2* with concomitant microRNA processing gene mutations, and chromosomal copy number alterations including 1q gain and loss of heterozygosity of both 1p and 16q [7,10,11,12].

In addition to these genetic features, a large Children’s Oncology Group case-cohort study identified that high telomerase RNA expression was associated with disease relapse in favorable histology WT [13]. Unfavorable histology WT was not evaluated by this study. Telomerase expression is found in over 85% of cancers, but is tightly repressed in most normal human tissues, illustrating the fundamental role it can play in cancer [14]. The telomerase holoenzyme ribonucleoprotein complex is responsible for adding repetitive nucleotide sequences (TTAGGG) to the ends of chromosomes to counteract progressive loss of telomeric DNA that occurs with replication and cell division. The enzyme complex consists of a protein catalytic subunit (TERT) and an RNA component (*TERC*) that serves as a template sequence for the deposition of nucleotide repeats. The rate-limiting step in telomerase activation is the transcription of *TERT* [15].

Multiple mechanisms of telomerase activation have been described in human malignancies including mutation, rearrangement, or methylation at the *TERT* promoter, *TERT* copy number gain, and upregulation of transcription factors which bind the *TERT* promoter region and increase transcription of *TERT* including N-MYC, a protein coded by the *MYCN* proto-oncogene [16,17,18]. Methylation of the *TERT* hypermethylated oncological region (THOR), a 433-base pair (bp) sequence upstream of the canonical *TERT* core promoter, is a frequent gain-of-function mechanism found in cancer [19]. The current study sought to define molecular and histologic correlates with *TERT* expression in Wilms tumor. The present study hypothesized that WT1 and N-MYC could regulate *TERT* expression in Wilms tumor because both proteins have been shown to interact with the *TERT* promoter and both are recurrently mutated or altered in Wilms tumor [6,7,18,20]. Improved understanding of telomerase biology in WT could lead to clinically relevant telomerase-based therapies. This study is novel because it identifies several mechanisms of *TERT* activation in Wilms tumor that could be of therapeutic interest in the future.

## 2. Materials and Methods

### 2.1. Establishment of Heterotopic WTPDX

Primary human WT samples were implanted into the flanks of CB17 *scid*^−/−^ mice (6–8 weeks old; Taconic Farms, Hudson, NY, USA) as previously described [20]. Xenograft tumor tissue from early passages (2–3) was snap frozen in liquid nitrogen for molecular studies or fixed in 10% neutral buffered formalin for histology.

### 2.2. DNA and RNA Analysis

Genomic DNA was extracted using a QiAamp DNA Mini kit (Qiagen, Hilden, Germany, RRID:SCR_008539) from 45 WTPDX, 39 available corresponding primary tumors and three normal adult kidney samples and was used for STR analysis, *TERT* promoter sequencing, and *TERT* promoter methylation analysis. In addition, total RNA was extracted from 37 paired primary tumors and WTPDX, eight additional WTPDX, and three normal kidney specimens by using the Qiagen RNeasy Midi kit (Qiagen). Commercially available pooled total RNA from four human fetal kidney specimens was also included (Takara, Kusatsu, Japan) and used for RNA-seq and quantitative real time PCR (qRT-PCR). This specific method for RNA extraction is also stated in a previous publication [20].

### 2.3. Authentication by STR Analysis

Genetic profiling analysis of WTPDX and cell lines in this study was performed by using 15 STR loci plus Amelogenin included in the PowerPlex^®^ 16 System kit (Promega, Madison, WI, USA). Electropherograms for all multiplex PCR-amplified products were obtained by using the 3730 xl DNA Analyzer (Applied Biosystems, Foster City, CA, USA) and were analyzed by using GeneMapper software (ThermoFisher Scientific, Waltham, MA, USA). This specific method for STR profiling is also stated in a previous publication [20].

### 2.4. TERT Promoter Sequencing

The *TERT* promoter region (chr5: 1,295,034–1,295,112; GRCh/hg38) was amplified by PCR using the following primers:$$\mathrm{Forward}-5^{\prime }\;\mathrm{CACAGCGCTGCCTGAAACTCG}\;3^{\prime }$$
$$\mathrm{Reverse}-5^{\prime }\mathrm{CCACGTGGCGGAGGGACTG}\;3^{\prime }$$

Sanger sequencing was performed (3730 xl DNA Analyzer, Applied Biosystems). Sanger sequencing electropherograms were manually reviewed for the presence of the canonical *TERT* promoter mutations C250T and C228T.

### 2.5. TERT Promoter Methylation Analysis

Genomic DNA (1 µg) from 39 WTPDX and corresponding primary tumors and three normal adult kidney specimens was bisulfite converted by using the EZ DNA Methylation kit (Zymo Research Corp, Irvine, CA) according to the manufacturer’s instructions. Converted samples were processed and hybridized to the Infinium MethylationEPIC BeadChip (850 K) system (Illumina, San Diego, CA, USA) according to published protocols [21]. The methylation score of each CpG site in the array is represented as a beta (β) value and was computed using the methylation module of the GenomeStudio software (version 1.9.0; Illumina). This specific method for 850 K MethylationEPIC beadchip analysis is also stated in a previous publication [20]. The beta values of three CpG sites (cg17166338, cg10767223, cg11625005) within the *TERT* promoter region were compared between WT and normal kidney.

### 2.6. TERT Promoter Bisulfite Sequencing

Genomic DNA was treated with sodium bisulfite using an EpiTect Bisulfite kit (Qiagen). Bisulfite-treated DNA was directly sequenced using primers corresponding to the core *TERT* promoter region. 18 CpG islands were analyzed in the targeted region of DNA amplification. Bisulfite conversion of greater than 50% of the CpG islands were defined as unmethylated for this analysis, less than 50% of the CpG islands were defined as hemimethylated, and no bisulfite conversion of any of the CpG islands were defined as fully methylated. Primers were designed using the MethPrimer tool [22] and primer sequences were:$$\mathrm{Forward}-5^{\prime }\;\mathrm{GGAAGTGTTGTAGGGAGGTATTT}\;3^{\prime }$$
$$\mathrm{Reverse}-5^{\prime }\;\mathrm{CATAATATAAAAACCCTAAAAACAAA}\;3^{\prime }$$

### 2.7. TERT Copy Number Analysis

Copy number alterations for the *TERT* locus were analyzed by Multiplex Ligation-dependent Probe Amplification (MLPA) using the SALSA P257 TERT-DKC1 kit (MRC Holland, Amsterdam, Netherlands). The GeneMapper Software (ThermoFisher Scientific) was used to perform DNA sizing and allele calls, and plots were generated and analyzed with Coffalyser software (MRC Holland). Reference samples [normal human kidney (*n* = 3; Amsbio, Abingdon, UK) and blood (*n* = 2; Human Genomic DNA, Promega, WI, USA)] from healthy individuals were included. This specific method for MLPA is also stated in a previous publication [20].

### 2.8. RNA Isolation and Quantitation of Gene Expression in Tumor Specimens or Xenografts

RNA-seq library preparation, sequencing, read mapping, and generation of gene level read counts and fragments per kilobase of exon per million mapped fragments (FPKM) values were generated as previously described [20]. For all RNA-seq-based analyses, FPKM values were transformed by log2(FPKM + 0.01). In addition, 1μg of total RNA was used for cDNA synthesis by using SuperScript IV VILO master mix kit (ThermoFisher, Grand Island, NY, USA). Fifty ng of cDNA was used for quantitative real time PCR (qRT-PCR) using Taqman technology (Applied Biosystems). This specific method is also stated in a previous publication [20]. The following Taqman probes designed for exon-exon boundaries (all cat#4331182) were utilized: *TERT* (assay ID Hs00972650_m1), *TERC* (Hs03454202_s1), *WT1* exons 3–4 (Hs01103751_m1), *WT1* exons 9–10 (Hs01103755_m1), and *MYCN* (Hs00232074_m1). Values were normalized by *ACTB* (Hs01060665_g1). Taqman-based qRT-PCR was performed using the Taqman Fast Advanced Master Mix (cat#4444557) according to the manufacturer’s protocol.

### 2.9. Quantification of In Vitro Gene Expression and Gene Set Enrichment Analysis

Raw RNA-seq FastQ files were input into the Workflow for the Analysis of RNA-seq Differential Expression (WARDEN) program using the St. Jude Cloud cloud-based analysis tool [23]. Using the WARDEN workflow, log2 counts per million (log2CPM) values were used to quantify gene expression according to mapped RNA-seq reads. The WARDEN workflow was also used to generate input files for gene set enrichment analysis (GSEA) [24]. GSEA software v4.0.3 (UC San Diego and Broad Institute) was used to analyze differential expression of Gene Ontology (GO) lists among samples of interest. Enrichment scores with associated statistical significance and adjustments for multiple hypothesis testing were made using GSEA software. The top 20 differentially regulated gene lists were displayed for each phenotype of interest. qRT-PCR was performed for in vitro experiments using Taqman technology as described above.

### 2.10. Telomerase Repeat Amplification Protocol (TRAP Assay)

Telomerase activity was determined using the TRAPeze telomerase detection kit (Millipore, Burlington, MA, USA), following the manufacturer’s protocol. In brief, frozen WT xenograft tissues were homogenized in CHAPS buffer and incubated at 25 °C for 40 min. For negative controls, lysates were heated up to 95 °C for 5 min to deactivate telomerase. Nucleotide templates for deposition of telomeric repeats and primers provided in the kit were added. The reaction was subjected to 29 PCR cycles at 95 °C for 30 s, 52 °C for 30 s, and 72 °C for 45 s. TRAP assay products were run on a 15% polyacrylamide gel and stained with SYBR green (ThermoFisher) for visual representation of telomerase activity and confirmation of positive and negative controls. Heat treatment (which inactivates telomerase) served as a negative control for each specimen. The human telomerase-positive control cell pellet provided in the TRAPeze kit (Millipore) served as a positive control for telomerase activity. TRAP assay products were quantified by capillary electrophoresis using 5′-fluorescent labeled (5′ 6-FAM) primers (IDT, Coralville, IA, USA) and calculating the total product generated (TPG). One TPG corresponds to the amount of telomerase activity detected in one immortal cell.

### 2.11. Western Blot for WT1, N-MYC, and Beta Actin

Total protein was extracted from the xenograft tumor samples using RIPA lysis buffer. Thirty ug of protein were separated by 4–20% gradient SDS polyacrylamide gel electrophoresis and transferred to nitrocellulose membrane using the iBlot2 transfer system (Life Technologies, Carlsbad, CA, USA). Membranes were blocked in 5% skim milk 50 mM Tris-buffered (pH 7.5) saline (TBS), containing 150 mM NaCl and 0.1% Tween 20 (TBST) and probed with WT1 monoclonal rabbit antibody raised against aa50–250 (dilution 1:1000, cat#ab89901, Abcam, Cambridge, MA, USA), N-MYC (B8.4B) N-terminal monoclonal mouse antibody (1:1000, cat#sc53993, Santa Cruz), N-MYC (D1V2A) C-terminal rabbit monoclonal antibody (1:1000, Cell Signaling Technology #84406) and β-actin monoclonal mouse antibody (1:4000, cat#MA5-15739, Invitrogen) or GAPDH (14C10) monoclonal rabbit antibody (1:1000, cat#21182, Cell Signaling Technology). After an incubation with horseradish peroxidase-conjugated anti-rabbit (1:5000, Thermo Fisher cat#A24537) or anti-mouse IgG (1:5000,Thermo Fisher cat#A24518), membranes were developed with HRP-detecting SuperSignal West Pico Chemiluminescent Substrate (Thermo Fisher) and were imaged using a Li-Cor Western blot imaging system (Li-cor, Lincoln, NE, USA).

### 2.12. Cell Line Acquisition and Validation

The human embryonic kidney HEK293 (RRID:CVCL_0045) and favorable histology WT PDM182 (HCM-BROD-0051-C64 (ATCC^®^PDM-182^™^) cell lines were acquired from American Type Culture Collection (ATCC; Manassas, VA). The anaplastic WT cell line 17.94 (RRID:CVCL_D704) was obtained from the European Collection of Authenticated Cell Cultures [25]. The anaplastic WiT49 cell line (RRID:CVCL_0583) was acquired from the laboratory of Dr. Herman Yeger [26]. The favorable histology WT COGW408 cell line was acquired from the Children’s Oncology Group/Alex’s Lemonade Stand Childhood Cancer Repository. The human acute promyelocytic leukemia suspension cell line NB4 (RRID:CVCL_0005) was acquired from the laboratory of Dr. Charles Mullighan [27]. All cell lines were tested for mycoplasma contamination and validated using short tandem repeat (STR) profiling monthly when in use.

### 2.13. shRNA Knockdown of WT1 in Human Wilms Tumor Cells

Twenty-four hours after plating, indicated cells were transduced with premade lentiviral particles at a multiplicity of infection (MOI) of 2 in the presence of 10 ug/mL polybrene (hexadimethrine bromide Sigma-Aldrich Cat#107689). The growth medium was replaced the morning after transduction. Forty-eight hours post-transduction, selection with puromycin was initiated for one week prior to use and maintained throughout all subsequent experiments.

### 2.14. CRISPR-Cas9-Mediated Deletion of WT1 in Human Wilms Tumor Cells

WT1^−/−^ cells were generated using CRISPR-Cas9 technology. Briefly, 200,000 HEK293 or WiT49 cells were transiently transfected with precomplexed ribonuclear proteins (RNPs) consisting of 100 pmol of chemically modified sgRNA (5′-GAGUAGCCCCGACUCUUGUA-3′, Synthego), 35 pmol of Cas9 protein (St. Jude Protein Production Core), and 500 ng of pMaxGFP (Lonza) via nucleofection (Lonza, 4D-Nucleofector™ X-unit) using solution P3 and program CM-130 (HEK293) or CA-137 (WiT49) in a small (20 μL) cuvette according to the manufacturer’s recommended protocol. Five days post nucleofection, cells were single-cell sorted by FACs to enrich for GFP+ (transfected) cells, clonally selected, and verified for the desired targeted modification via targeted deep sequencing using gene specific primers with partial Illumina adapter overhangs as previously described [28]:$$\mathrm{CAGE}74.\mathrm{DS}.F-5^{\prime }\;\mathrm{GGTCTGCACCTGCCACCCCTTCTTT}-3^{\prime }$$
$$\mathrm{CAGE}74.\mathrm{DS}.R-5^{\prime }\;\mathrm{GTTTGCCCAAGACTGGACAGCGGGC}-3^{\prime }\;\left(\mathrm{overhangs}\;\mathrm{not}\;\mathrm{shown}\right)$$

Briefly, cell pellets of approximately 10,000 cells were lysed and used to generate gene specific amplicons with partial Illumina adapters in PCR#1. Amplicons were indexed in PCR#2 and pooled with targeted amplicons from other loci to create sequence diversity. Additionally, 10% PhiX Sequencing Control V3 (Illumina) was added to the pooled amplicon library prior to running the sample on an Miseq Sequencer System (Illumina) to generate paired 2 × 250 bp reads. Samples were demultiplexed using the index sequences, fastq files were generated, and NGS analysis of clones was performed using CRIS.py [29]. Knockout clones containing only out-of-frame indels were identified. Final clones were authenticated using the PowerPlex^®^ Fusion System (Promega) performed at the Hartwell Center (St. Jude) and tested negative for mycoplasma using the MycoAlert^TM^Plus Mycoplasma Detection Kit (Lonza). *WT1* knockouts were confirmed by western blot, RNA-seq, and qRT-PCR. These specific methods are common to studies performed by the St. Jude Children’s Research Hospital Center for Advanced Genome Engineering and therefore may be partly included in other manuscripts that utilized this shared institutional resource.

Two validated WT1 gRNAs were used in this study. The sequences of gRNAs are:$$\mathrm{CAGE}74.\mathrm{WT}1.g2\;5^{\prime }-\mathrm{GAGTAGCCCCGACTCTTGTANGG}-3^{\prime }$$
$$\mathrm{CAGE}74.\mathrm{WT}1.g3\;5^{\prime }-\mathrm{AGCCCCGACTCTTGTACGGTNGG}-3^{\prime }$$


### 2.15. Crystal Violet Staining

Cells were plated at 3000 cells per well and grown for 3 weeks. To fix and stain cells, media was aspirated, and cells were washed with Dulbecco’s phosphate buffered saline without calcium or magnesium (DPBS, Lonza) and fixed with 4% formaldehyde in PBS (PFA) for 20 min. Once PFA was removed, cells were stained with 0.1% crystal violet stain for 1 h.

### 2.16. TERT Promoter Activity Luciferase Assay

*TERT* promoter luciferase plasmids were obtained from Addgene (pGL4.0-TERT WT #84924, pGL4.0-TERT G250A #82925, pGL4.0-TERT G228A #82926) [30]. WiT49 and HEK293 cells were seeded at a density of 30,000 cells/mL in a 96-well plate (PerkinElmer CulturPlate 96 #6005680). Twenty-four hours post-seeding, cells were transfected with 90 ng vector, 9 ng pGL4.74 (*Renilla* control), and the Lipofectamine 3000 transfection reagent system (ThermoFisher Cat# L3000008). At 48 h post-transfection, firefly luciferase activity was measured by the Dual-Glo Luciferase Assay system (Promega cat #E2920) and normalized against *Renilla* luciferase activity. All experiments were performed with four replicate wells. For *TERT* promoter luciferase assays involving N-MYC overexpression, a longer *TERT* promoter sequence (GRCh37/hg19 chr5:1,295,105–1,296,183) was subcloned into the pGL4.0-TERT WT#84294 plasmid backbone (Genscript Biotech) to include two E box motifs which constitute N-MYC binding sites.

### 2.17. Inducible Wt1 Knockout Mouse Model

Mice bearing the *Wt1* null allele (*Wt1*^+/−^), *Wt1*-floxed allele *Wt1*^fl/+^, and the tamoxifen inducible CAGG-*Cre-ER*^TM/+^ allele have been previously described and were obtained from the laboratory of Dr. Vicki Huff [31,32,33]. Heterozygous *Wt1*^+/−^ mice were crossed with tamoxifen-inducible *Cre-ER*^TM^ mice to create mice bearing the *Wt1*^+/−^-*Cre-ER*^TM^ genotype. Heterozygous mice bearing one floxed and one null Wt1 allele (*Wt1^fl/^*^−)^ were crossed with *Wt1*^+/−^-*Cre-ER*^TM^ mice. Formation of a vaginal plug was defined as embryonic day E0.5. Pregnant dams were treated with 5 mg/40 g body weight intraperitoneal tamoxifen at E11.5 to activate Cre-mediated recombination and knock out *Wt1*. Embryos were harvested at E15.5. Embryonic kidneys were procured using a dissecting microscope and immediately snap frozen in liquid nitrogen. RNA was isolated from embryonic kidneys and qRT-PCR was performed using Taqman probes (Applied Biosystems) specific for murine *Wt1*, *Tert*, and *ACTB* using techniques described above.

### 2.18. GUDMAP Database

The relationship between Wt1 and Tert expression in the murine embryonic kidney was further investigated using data from the GenitoUrinary Development Molecular Anatomy Project (GUDMAP) [34,35]. Whole mount RNA in situ hybridization imaging data was queried in e15 murine embryonic kidneys for Tert (RID N-H6X0; probe ID N-FWGY), Wt1 (RID N-H0RE; probe ID N-FPSW), Six2 (positive control; RID N-GZX4; probe ID N-FVHT) and Bdnf (brain-derived neurotropic factor; negative control; RID N-H3X6; probe ID N-FXH8) [36].

### 2.19. N-MYC Overexpression

N-MYC wildtype and P44L mutant N-MYC overexpression was achieved by lentiviral pCL20-loxp-MYCN-iGFP and pCL20-loxp-MYCN-P44L-iGFP plasmids from Dr. Brian Sorrentino’s laboratory at St Jude Children’s Research Hospital. Cells expressing GFP were sorted using flow cytometry within the Flow Cytometry and Cell Sorting Shared Resource and expanded prior to experimental use for one week.

### 2.20. Lentiviral Production

Transfecting HEK93T cells with viral vectors was achieved by combining 6 ug of target vector, 3 ug of CAG-kGP1-1R, 1 ug of CAG4-RTR2, and 1 ug of CAG-VSV-G plasmids in 400 μL of DMEM without serum or L-glutamine. PEIpro transfection reagent (Polyplus 115-010) was added at 2:1 (PEIpro μL: ug of plasmid) per 100 mm dish of cells, mixed well, and incubated at RT for at least 20 min, prior to adding cells. The following day, fresh medium was added to cells. For 3–4 days, viral media was harvested and replaced twice per day. Viral media was centrifuged at 1500 RPM for 10 min and filtered through a 0.45 um vacuum filter. Virus was concentrated by ultracentrifugation at 28.5 kRPM for 2 h at 4 °C, aspirated, and resuspended in either OptiMEM or PBS, aliquoted, and was frozen at −80 °C until use.

### 2.21. Statistics

Data were tested for normal distribution using the D’Agostino-Pearson normality test. For data with a normal distribution, continuous variables were compared using the two-tailed *t*-test. For data determined not to adhere to a normal distribution, continuous variables were compared using the non-parametric Mann-Whitney test. Correlations between continuous variables were determined by calculating the Spearman correlation coefficient because these data sets had non-normal distributions. A linear regression was performed to display the relationship between percent blastema and telomerase activity. Tumor biospecimen results are displayed with individual data points shown and the median represented as a line. In vitro experiments are shown as bar graphs with the bar representing the mean and the error bars indicating the standard deviation. Statistics were performed using GraphPad Prism software (v 9.0). A *p*-value of <0.05 was considered statistically significant for all tests.

## 3. Results

### 3.1. WT Contain TERT Promoter Mutations, Promoter Hypermethylation, and TERT Locus Amplification

To confirm tumor-specific upregulation of *TERT* in Wilms tumor, the present study compared *TERT* expression in WT to normal kidney and developing kidney. Primary WT and corresponding xenografts were found to have significantly higher expression of *TERT* by RNA-seq than fetal kidney (*p* = 0.005) and normal kidney samples (*p* = 0.0006; Figure 1a). This study then compared the *TERT* expression in primary Wilms tumors to other pediatric solid tumors using data from the St. Jude Genome Project for reference (Figure 1b) [23]. Median expression of *TERT* in Wilms tumor was found to be between the two modes of *TERT* expression seen in neuroblastoma and medulloblastoma, which have bimodal *TERT* expression according to the tumor molecular subtype (Figure 1b) [18,37].

Next, this study sought to identify the types of *TERT* alterations in Wilms tumor specimens. Canonical activating mutations C250T (KT-33 and KT-74) and C228T (KT-71) were identified in the *TERT* promoter by Sanger sequencing in 3/45 (6.7%) snap frozen WT specimens and corresponding patient-derived xenografts (Figure 1c). KT-33 and KT-74 tumors were from patients who went on to experience disease relapse and KT-71 was from a patient with SIOP-high-risk histology who had identification of diffuse anaplasia after neoadjuvant chemotherapy Appendix A). To validate the presence of *TERT* promoter mutations in a separate cohort, we queried the NCI TARGET data for Wilms tumors that underwent whole genome sequencing and found that 3/81 (3.7%) contained the C228T *TERT* promoter mutation (Appendix A) [7]. These three *TERT* promoter mutant primary tumor specimens were found to have significantly increased expression of *TERT* relative to other tumors in the NCI TARGET Wilms tumor RNA-seq data (Appendix A, Figure 1d). However, in the xenograft data set, the three *TERT* promoter mutant specimens were not found to have significantly higher *TERT* expression or telomerase activity than non-mutant specimens. *TERT* promoter luciferase assays documented increased *TERT* promoter activity in the presence of the C228T and C250T mutations when compared to the wild-type promoter sequence in anaplastic Wilms tumor WiT49 cells and transformed human embryonic kidney HEK293 cells (Figure 1e).

Near-ubiquitous hypermethylation of three CpG islands (cg10767223, cg11625005, and cg171663338), one of which (cg11625005) is contained within the *TERT* hypermethylated oncological region (THOR) located upstream of the core *TERT* promoter [19], was seen in WT and corresponding patient-derived xenografts when compared to normal kidney samples (Figure 1f). These observations were extended by performing bisulfite sequencing of the core *TERT* promoter region. Twenty-one out of 45 (46.7%) xenografts exhibited hemimethylation of the core *TERT* promoter region, 23 (51.1%) were fully methylated, and one xenograft (KT-25, 2.2%) exhibited no methylation in this region (Appendix A). Of note, all three of the xenografts found to have canonical *TERT* promoter mutations (KT-33, 74, 71) exhibited hemimethylation of the *TERT* promoter region analyzed by bisulfite sequencing analysis (Appendix A). Copy number gain at the *TERT* locus was identified by Multiplex-ligation dependent probe amplification (MLPA) in three specimens (KT-28, 31, 43; Appendix A); of note KT-28 was also found to have combined loss of heterozygosity (LOH) of 1p/16q, and KT-31 and 43 were also found to have copy number gain at 1q, both of which are clinically validated poor prognostic indicators in WT (Appendix A). Among the six samples with *TERT* promoter mutations or copy number gain, all six exhibited some aspect of adverse biology or clinical behavior including disease relapse, diffuse anaplasia, 1q gain, or combined loss of heterozygosity of 1p or 16q. In contrast, *TERT* promoter hypermethylation extended beyond specimens with adverse biology and was found throughout the spectrum of Wilms tumor (Appendix A). Overall, these data show that a small proportion of Wilms tumor samples have high *TERT* expression. These data suggest three ways by which *TERT* gain-of-function is achieved via direct genetic or epigenetic mechanisms in Wilms tumor: canonical gain-of-function promoter mutations, *TERT* locus amplification, and promoter hypermethylation.

### 3.2. WT1-Mutant Wilms Tumors Exhibit Lower TERT Expression and Telomerase Activity

The present study explored the association between *WT1* and *TERT* because *WT1* is mutated in approximately 20% of Wilms tumor specimens and has been previously shown to interact with the *TERT* promoter [38]. Biallelic inactivating mutations were identified in 7/45 (15.6%) Wilms tumor xenograft specimens and corresponding primary tumors (Appendix A). As expected, *WT1*-mutant xenografts had significantly lower *WT1* expression by qRT-PCR (*p* = 0.0004; Figure 2a) when compared to *WT1* wild-type xenografts. These *WT1*-mutant xenografts were confirmed to have absent or low WT1 by western blot (Appendix A). *WT1*-mutant xenografts had significantly lower expression of *TERT* by qRT-PCR (*p* = 0.00766; Figure 2b) than *WT1* wild-type xenografts. *WT1*-mutant xenografts had significantly lower telomerase activity detected by the quantitative telomerase repeat amplification protocol (TRAP) assay (*p* = 0.0128; Figure 2c). A statistically significant positive correlation was observed between *TERT* expression and *WT1* expression by qRT-PCR (Spearman r = 0.47; *p* = 0.001; Figure 2d). These relationships were confirmed by RNA-seq data (Appendix A). These data suggest a coupling of WT1 function and *TERT* expression in Wilms tumor cells.

### 3.3. WT with Unfavorable Histology Exhibit Higher TERT Expression and Telomerase Activity

Six tumors with diffuse anaplasia were compared to 39 tumors with favorable histology and found to have a significant increase in *TERT* expression by qRT-PCR (*p* = 0.0062; Figure 3a. This relationship was confirmed using RNA-seq data (Appendix A). Tumor xenografts were found to have significantly higher telomerase activity than favorable histology xenografts (*p* = 0.0021; Figure 3b). A significant positive correlation was found between the percentage of blastema in xenograft samples telomerase activity (Spearman r = 0.424, *p* = 0.004; Figure 3c). These data suggest that high-risk Wilms tumor with diffuse anaplasia have higher *TERT* expression than other Wilms tumors and that the proportion of blastema in a Wilms tumor correlates with the level of *TERT* expression.

### 3.4. Effects of WT1 Knockdown and Knockout on Wilms Tumor Cells In Vitro

The positive correlation of *WT1* and *TERT* expression suggests that WT1 may be involved in regulation of *TERT*. The human WT cell lines WiT49, 17.94, COG-W-408, and PDM-182 were screened for *WT1* expression using qRT-PCR and western blot and only the anaplastic WT cell line WiT49 exhibited detectable *WT1* expression by both modalities (Appendix A). Therefore, WiT49 was utilized for further experiments. Stable WiT49 cell lines with knockdown of *WT1* were developed using lentiviral-mediated transduction of shRNAs against *WT1* followed by puromycin selection. Two of four shRNAs (1114 and 0596 henceforth called shRNA1 and shRNA5, respectively) demonstrated adequate knockdown of WT1 by western blot when compared to control vector containing a non-targeting control shRNA (Figure 4a). shRNA-mediated knockdown of *WT1* was associated with reduction in colony formation (Figure 4b), reduction in cellular proliferation (*p* < 0.001; Figure 4c), and decreased expression of *TERT* by qRT-PCR (Figure 4d). However, a nonsignificant reduction in telomerase activity was observed with shRNA knockdown of *WT1* (Figure 4e). Because other Wilms tumor cell lines with expression of *WT1* were unavailable, the NB4 acute promyelocytic leukemia cell line, which demonstrates high levels of *WT1* expression, was used to validate the association between shRNA knockdown of *WT1* and decreased *TERT* expression (Appendix A). Using CRISPR-Cas9-mediated disruption of the second and third of four DNA-binding zinc-finger domains of *WT1*, WT1-null WiT49 cells (WiT49-1D9 and WiT49-1G11) and WT1-null HEK293 cells (HEK293-1E3 and HEK293-3H5) were created. All clones were confirmed to have frameshift mutations in exon 7 of WT1 by whole exome sequencing analysis (Table 1). WT1 knockout was validated on the protein level by western blot (Figure 4f). All clones demonstrated a statistically significant reduction in *WT1* expression (Figure 4g). WiT49-1D9, WiT49-IG11, and HEK293-3H5 *WT1*-knockout cells demonstrated a statistically significant decrease in *TERT* expression (Figure 4g). The HEK293 cell line showed a very low baseline expression level of *TERT* (Figure 4g).

Changes in gene expression in WiT49 and HEK293 *WT1* knockout clones compared to parental controls were evaluated in an unbiased manner by RNA-seq and gene set enrichment analysis of 2789 Gene Ontology (GO) Pathways. Significant differences were identified in the gene sets downregulated between the *WT1* knockout clones (WiT49-1D9 and WiT49 1G11) compared to parental control cells. For WiT49-1D9, the GO_TELOMERE_MAINTENANCE_VIA_TELOMERE_LENGTHENING pathway was the 5th most downregulated gene set (Appendix A) and telomere-related pathways (GO_PROTEIN_LOCALIZATION_TO_CHROMOSOME_TELOMERIC_REGION, GO_ESTABLISHMENT_OF_PROTEIN_LOCALIZATION_TO_TELOMERE) also constituted the 10th and 20th most downregulated gene sets, respectively (Appendix A). In contrast, the WiT49-1G11 clone had no telomere-relevant pathways identified in the top 20 most downregulated gene sets (Appendix A). The WiT49-1D9 and WiT49-1G11 clones were then compared directly using gene set enrichment analysis. Four of the top 20 Gene Ontology Pathways enriched in the WiT49-1G11 clone when compared to the WiT49-1D9 clone were found to be telomere-related including GO_PROTEIN_LOCALIZATION_TO_CHROMOSOME_TELOMERIC_REGION, GO_TELOMERASE HOLOENZYME COMPLEX, GO_ESTABLISHMENT_OF_PROTEIN_LOCALIZATION_TO_TELOMERE, and GO_TELOMERASE_RNA_BINDING (Appendix A). RNA-seq analysis of *WT1* comparing the WIT49-1D9 and WiT49-1G11 clones revealed that the expression of *TERT* was significantly downregulated in the WiT49-1D9 clone compared to the 1G11 clone (Figure 4g). RNA-seq reads for *WT1* showed peaks consistent with maintained expression of *WT1* exons 1–5 in the WiT49-1G11 clone, but not the WiT49-1D9 clone (Appendix A). To confirm this observation, qRT-PCR was performed using Taqman probes targeted to the *WT1* exon 3–4 junction and the *WT1* exon 9–10 junction. *WT1* exon 3–4 RNA was detected in WiT49-1G11 cells, but not in WiT49-1D9 cells. In contrast, the probe targeted to the exon 9-10 junction showed knockdown in both WiT49-1D9 and WiT49-1G11 cells (Appendix A). We therefore hypothesized that the WT1-1G11 clone maintains expression of the N-terminal portion of WT1 coded by exons 1-5. The predicted molecular weight of such a fragment is between 27.7 kDa and 35.4 kDa depending on the *WT1* isoform being expressed (presence or absence of alternatively spliced exon 5). Supporting this possibility, western blot analysis of WT1 identified a 30 kDa protein only in WiT49-1G11 cells, consistent with a WT1 N-terminal fragment (Appendix A). Telomere-related gene sets were not among the top 20 downregulated pathways in HEK293 WT1 knockout cells (HEK293-1E3 and HEK293-3H5; Appendix A); however, this is in the context of an extremely low baseline level of *TERT* expression in the HEK293 cell line (Figure 4G). These data suggest that *WT1* knockout is associated with reduction in telomere-related pathways in Wilms tumor cells, but not HEK293 human embryonic kidney cells.

### 3.5. Wt1, Tert, and Kidney Development

Because Wilms tumor is characterized by overexpression of genes important in kidney development, particularly those highly expressed in the pre-induction metanephric mesenchyme (including *WT1*), the current study analyzed the role of *TERT* in kidney development and the relationship of *WT1* and *TERT* in this context. Heterozygous mice bearing one *Wt1* null and one floxed *Wt1* allele (*Wt*^fl/^^−^) were crossed with *Wt1*^+/^^−^-*Cre-ER*^TM^ mice. Pregnant dams were treated with 5 mg/40 g body weight intraperitoneal tamoxifen at E11.5 to activate Cre-mediated recombination and knock out of *Wt1*
Appendix A). Embryos were harvested at E15.5 and RNA was isolated from embryonic kidneys. No decrease in *Tert* expression was observed in Wt1^−/−^ or CAGG-CreER^TM^-Wt1^fl/^^−^ embryonic kidneys when compared to embryonic kidneys with competent *Wt1* expression (Figure 5a).

To validate this observation using a different dataset, whole mount RNA in situ hybridization data in e15 murine embryonic kidneys available via the GUDMAP database was queried [34,35]. *Tert* was not found to be expressed at significant levels in the murine embryonic kidney (Figure 5b). These data suggest that Tert is not essential to normal kidney development and Wt1 alone is not enough to induce *Tert* expression, at least in murine kidney development.

### 3.6. MYCN and TERT Expression in Wilms Tumor

*MYCN* amplification and mutation (P44L) are among the most frequent recurrent genetic alterations in anaplastic Wilms tumor. Furthermore, N-MYC is known to regulate *TERT* in other cancer types, including neuroblastoma [6,18]. Therefore, we hypothesized N-MYC regulates *TERT* in Wilms tumor. *TERT* expression was compared between xenografts with *MYCN* amplification (KT-53, 48, 27, 39) and *MYCN* P44L mutation (KT-81) to those with wild type, diploid *MYCN* status and found nonsignificant higher *TERT* expression in this data set (Figure 6a). This observation did not achieve statistical significance possibly because of the low number of samples in the *MYCN* mutant or amplified group. Therefore, the NCI-TARGET data (which contains a much greater number of specimens with *MYCN* amplification due to enrichment for tumors with diffuse anaplasia or from patients who experienced disease relapse) was queried and found significantly higher *TERT* expression in *MYCN* amplified or P44L mutant specimens when compared to *MYCN* diploid, wild-type specimens by RNA-seq (Figure 6b). Overexpression of N-MYC in WiT49 cells (Figure 6c) caused increased *TERT* promoter luciferase activity (Figure 6d) and *TERT* expression (Figure 6e). Overexpression of the P44L mutant N-MYC (most common *MYCN* mutation found in Wilms tumor; Figure 6c) was associated with increased *TERT* promoter luciferase activity (Figure 6d) and *TERT* expression (Figure 6e) relative to wild type *MYCN*. Of note, the P44L mutation disrupts the epitope recognized by the N-MYC (N-terminal) antibody but is detected by the alternate N-MYC (C-terminal) antibody (Figure 6c). These data suggest that N-MYC regulates *TERT* expression in Wilms tumor and that common *MYCN* alterations in WT result in increased *TERT* expression.

## 4. Discussion

This study begins to elucidate potential mechanisms of telomerase activation in Wilms tumor, which has been previously associated with disease-relapse in a large case-cohort study conducted by the Children’s Oncology Group [13]. The possible mechanisms are comprised of the canonical activating *TERT* promoter mutations C228T and C250T, hypermethylation of the *TERT* hypermethylated oncological region (THOR) [19] and adjacent sequences upstream of the core *TERT* promoter, copy number amplification of the *TERT* locus, and facilitation of *TERT* transcription by WT1 and N-MYC (Figure 7). This study shows that *TERT* transcription and telomerase activity are upregulated in patient-derived Wilms tumor xenograft specimens with diffuse anaplasia and are positively correlated with the percentage of blastema in WT specimens. Furthermore, this study shows a relationship between biallelic inactivating mutations in *WT1* and decreased *TERT* expression and telomerase activity. This relationship between *WT1* and *TERT* detected in biospecimens was corroborated by in vitro data showing that knockdown and knockout of WT1 expression is associated with decreased *TERT* transcription. The lack of a relationship between *Wt1* and *Tert* in murine embryonic kidney development suggests that gain of telomerase function is a later event in Wilms tumorigenesis and not a consequence of its hypothesized developmental origin.

In apparent contrast to the current study, WT1 has been shown previously to bind the *TERT* promoter and exert an inhibitory effect on *TERT* transcription and therefore telomerase enzyme activity [38]. These studies were performed by identifying a potential WT1 binding site in the *TERT* promoter region. Deletion of this consensus site impaired WT1 binding and consequently de-repressed *TERT* promoter activity using firefly luciferase-promoter constructs. However, this finding was shown in 293(T) cells, but not malignant HeLa cells indicating that the phenomenon could be cell-type dependent. The WT1 binding sequence in the *TERT* (chromosome 5p15.33) promoter “AGCGCCCGCGCGGGCGGG” reported by this study does not localize to the *TERT* promoter region when input into NCBI Nucleotide blast and therefore may not reflect the endogenous human *TERT* promoter. Furthermore, the effect of endogenous WT1 knockdown was not evaluated in this study, likely due to technological limitations at the time [38]. In contrast to these findings, the present study showed that knockdown of endogenous WT1 in WiT49 human Wilms tumor cells was associated with decreased *TERT* transcription.

This study found an association between *TERT* expression/telomerase activity and diffuse anaplasia, the most important prognostic factor in Wilms tumor. The previous Children’s Oncology Group analysis of *TERT* expression and disease relapse was limited to favorable histology, non-anaplastic Wilms tumors [13]. This study did not find an association between telomerase activity detected by the TRAP assay and the relative risk of disease relapse. However, this was perhaps because of the heterogeneous way in which the specimens were processed by contributing centers before submission to the biopathology center with accompanied degradation of telomerase activity. Diffuse anaplasia, defined by the histologic presence of large hyperchromatic nuclei with abnormal multipolar mitoses, is the single most important poor prognostic indicator in Wilms tumor, necessitating intensified chemotherapy and radiation for stage II-IV disease [1]. Diffuse anaplasia is thought to be caused by acquisition and expansion of a *TP53* mutant clone in a previously favorable histology Wilms tumor [3]. Inactivation of p53 and activation of telomerase are frequently found in human cancers and may cooperate to cause cancer cell immortalization [39]. P53 deficiency has been shown to rescue the adverse effects of telomere loss and telomere-induced stress in cancer cells [40]. Furthermore, loss of p53 and transgenic expression of *Tert* were shown to result in an increased incidence of neoplasia in a mouse model [41]. Therefore, p53 mutations and gain-of-function alterations in *TERT* could play a cooperative role in anaplastic Wilms tumor pathogenesis.

This study shows that common *MYCN* alterations in WT including overexpression and P44L mutation are associated with increased *TERT* transcription. Elevated *TERT* expression is strongly associated with *TERT* promoter rearrangements or *MYCN*-amplification in neuroblastoma [18]. In the current study, we found an association between *MYCN* amplification and the P44L *MYCN* mutation and increased *TERT* expression in Wilms tumor xenografts and primary tumor specimens. The present study also showed that ectopic overexpression of the N-MYC P44L mutant is associated with increased *TERT* promoter activity and gene expression relative to wild type N-MYC and empty vector control cells. MYC proteins are known to bind E box motifs in the proximal *TERT* promoter near the transcription start site to facilitate *TERT* transcription [17]. High *MYCN* expression has also been associated with high-risk Wilms tumor as defined by post-treatment anaplasia or blastemal histology WT [6,42]. In summary, this study suggests that N-MYC participates in the *TERT* regulatory axis, which may govern telomerase function in Wilms tumor and be of therapeutic interest in future studies.

Wilms tumor is hypothesized to be a renal developmental malignancy, initiated by a blockade in stem cell differentiation that, with acquisition of additional genetic “hits”, results in embryonal tumorigenesis [43]. Consistent with this hypothesis, Wilms tumors were found to have consistently high expression of genes associated with the pre-induction metanephric mesenchyme in kidney development [7]. Furthermore, alterations of critical renal developmental pathway genes including *SIX1/2*, *CTNNB1*, *WT1*, and microRNA processing genes are suspected to be drivers of Wilms tumorigenesis [43,44]. The current study shows that *Tert* is not significantly expressed during murine embryonic kidney development and that there is no relationship between inducible *Wt1* knockout and bulk RNA *Tert* expression levels in murine nephrogenesis. These findings suggest that activation of *TERT* transcription in Wilms tumor is an acquired event later in tumor progression rather than a maintained characteristic of the renal stem cell population from which these tumors originate.

This study has important limitations. First, despite widespread WT1 detection in human Wilms tumor and associated patient-derived xenografts, *WT1* expression was detected in only one Wilms tumor cell line (WiT49) in the current analysis. The current study attempted to overcome this limitation by confirming the association between *WT1* knockdown and reduced *TERT* expression in NB4, an acute promyelocytic leukemia cell line with known high levels of *WT1* expression and HEK293 cells, a transformed human embryonic kidney cell line. Wilms tumor cells are notoriously difficult to establish as immortalized cell cultures and 2D cultures may more efficiently capture epithelial cells rather than blastemal cells that are characterized by high expression of *WT1* [45]. Furthermore, the current analysis includes a large library of well-characterized patient-derived xenografts that reflect the biologic heterogeneity of Wilms tumor and that have been shown to efficiently foster and propagate blastemal cells across passages [20]. These xenograft samples were all processed identically and immediately snap frozen upon procurement from recipient mice, which possibly results in more consistent telomerase activity data when compared to heterogeneously processed clinical specimens. Although the current analysis was conducted on early-passage untreated xenograft materials, it did not control for neoadjuvant chemotherapy treatment received by patients prior to surgical resection and thus the long-term effect of chemotherapy treatment on *TERT* expression or telomerase activity cannot be excluded. Finally, WT1 gain-of-function studies to determine effects on *TERT* and *MYCN* were not possible due to ectopic overexpression of WT1 causing cell death when compared to empty vector control cells (data not shown). Although this finding represents a limitation of the current study, it illustrates the paradoxical role of WT1 as a tumor suppressor and potential oncogene depending on its expression level and/or the cellular context and will be the subject of future study [46].

## 5. Conclusions

In conclusion, this study is the first to report multiple gain-of-function alterations in *TERT* in Wilms tumor including promoter mutations, promoter hypermethylation, and *TERT* locus amplification. Therefore, this work provides molecular mechanisms that support the previous observation that increased *TERT* RNA levels correlate with disease relapse in Wilms tumor. *TERT* levels were found to be higher in anaplastic Wilms tumor and much lower in the *WT1*-mutant molecular subgroup. Knockdown and knockout of *WT1* reduced *TERT* transcription and N-MYC overexpression increased *TERT* transcription. Telomere-targeted therapies may be a future treatment direction for anaplastic Wilms tumor and preclinical studies are warranted.

## Figures and Tables

**Figure 1 cancers-14-01655-f001:**
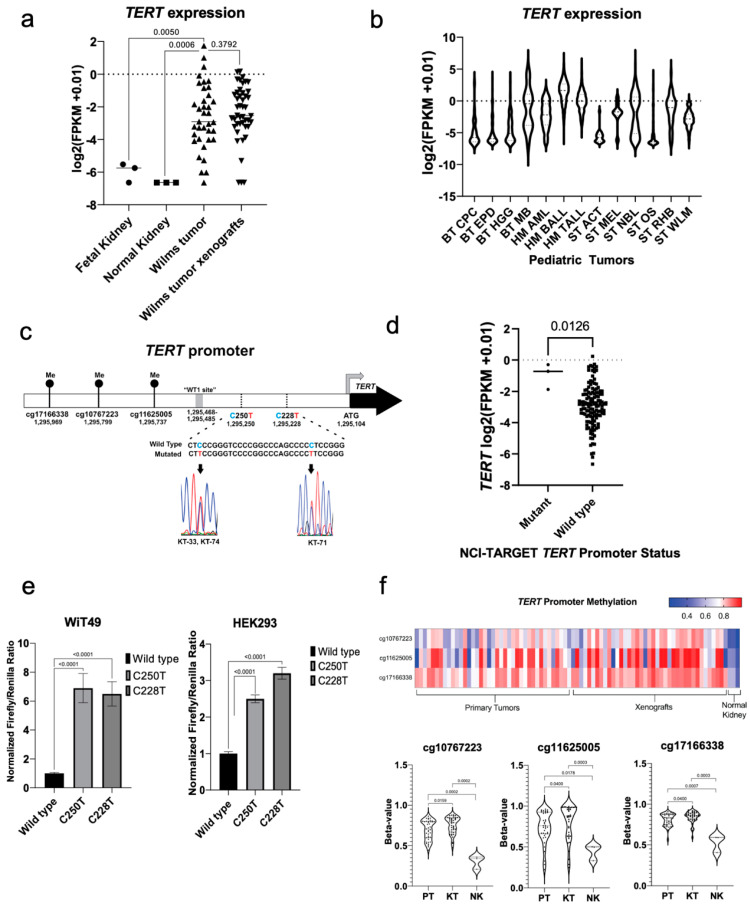
Mechanisms of *TERT* activation in Wilms tumor. (**a**) Wilms tumors have increased *TERT* expression compared to pooled fetal kidney mRNA (Mann-Whitney test *p* = 0.0050) and normal kidney (*p* = 0.0006) by RNA-seq, but similar *TERT* expression when compared to associated patient-derived xenografts (*p* = 0.3792). (**b**) RNA-seq *TERT* expression in Wilms tumor versus other pediatric solid tumors (BT—brain tumor, CPC—choroid plexus carcinoma, EPD—ependymoma, HGG—high grade glioma, MB—medulloblastoma, HM—hematologic malignancy, AML—acute myelogenous leukemia, BALL—B-cell acute lymphoblastic leukemia, TALL—T-cell acute lymphoblastic leukemia, ST—solid tumor, ACT—adrenocortical tumor, MEL—melanoma, NBL—neuroblastoma, OS—osteosarcoma, RHB—rhabdomyosarcoma, WLM—Wilms tumor). (**c**) The canonical *TERT* promoter mutations C250T and C228T are detected in 3/45 (6.7%) of Wilms tumor specimens and corresponding patient-derived xenografts. The map displays the *TERT* promoter region, the location of the putative WT1 binding site, and three CpG islands upstream of the *TERT* promoter, one of which (cg11625005) is in the *TERT* hypermethylated oncologic region (THOR). (**d**) The three C228T mutant patient samples from the NCI-TARGET Wilms tumor data set have significantly higher expression of *TERT* by RNA-seq when compared to wild type samples (*p* = 0.0126). (**e**) Transfection of WiT49 cells and HEK293 cells with *TERT* promoter luciferase plasmids containing the wild type, C250T, and C228T promoter mutant sequences confirmed increased *TERT* promoter activity (all two-sided *t*-test *p* < 0.0001). (**f**) Wilms tumors and associated patient-derived xenografts are found to have hypermethylation of CpG probes located at the THOR region and adjacent upstream sequences when compared to normal kidney specimens. Statistically significant increases in methylation were observed in xenografts (KT) compared to primary tumors (PT), but overall methylation patterns were quite similar according to violin plots and were both notably increased compared to normal kidney specimens (NK).

**Figure 2 cancers-14-01655-f002:**
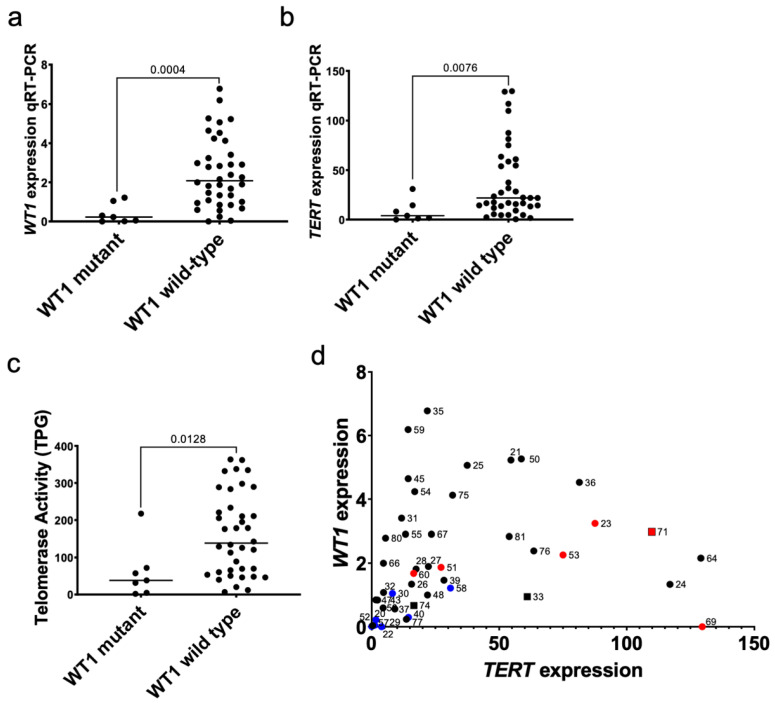
Relationship between biallelic inactivating mutations in *WT1* and *TERT* expression. (**a**) The expression of *WT1* was validated to be lower in specimens with biallelic inactivating mutations in *WT1* compared to wild type specimens by qRT-PCR (Mann-Whitney *p* = 0.0004). (**b**) *TERT* expression is significantly lower in *WT1*-mutant Wilms tumor patient-derived xenografts compared to wild type specimens by qRT-PCR (*p* = 0.0076). (**c**) The telomerase repeat amplification protocol (TRAP) assay was quantified to calculate the total product generated (TPG) and *WT1*-mutant specimens had significantly lower telomerase activity than *WT1*-wild type specimens (*p* = 0.0128). (**d**) A significant positive correlation between qRT-PCR quantification of *WT1* and *TERT* levels (Spearman r = 0.47; *p* = 0.001) was detected. Blue = *WT1* mutant xenografts, Red = anaplastic xenografts. Squares = xenografts with *TERT* promoter mutations. Circles = xenografts without *TERT* promoter mutations.

**Figure 3 cancers-14-01655-f003:**
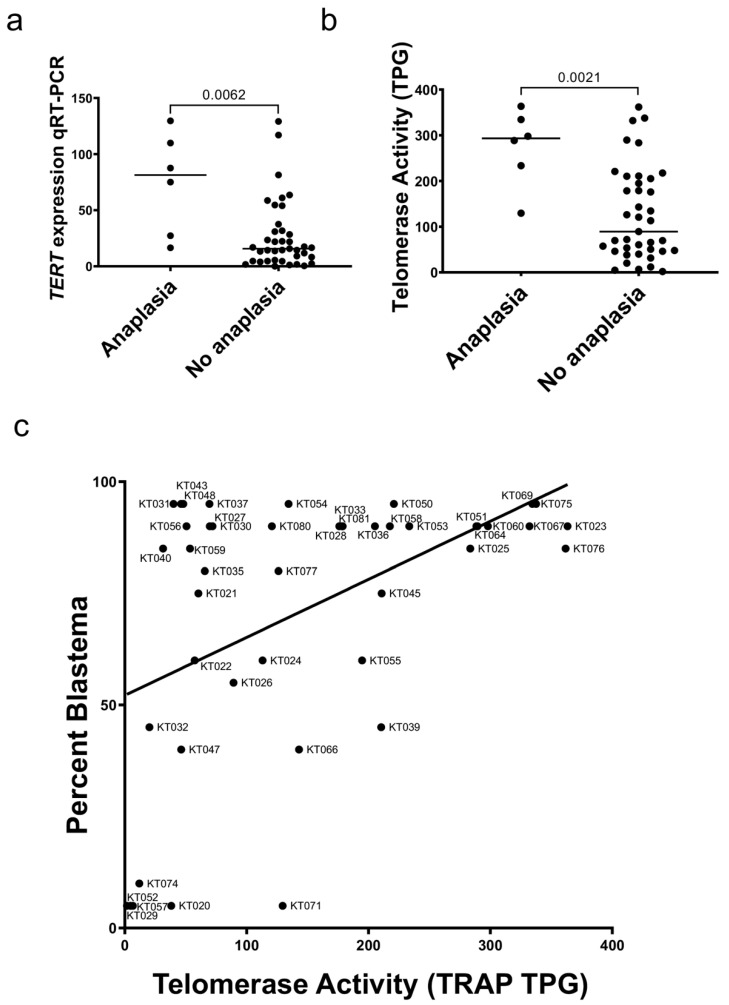
*TERT* and Wilms tumor histology. (**a**) Wilms tumor patient-derived xenografts with diffuse anaplasia had higher *TERT* expression versus favorable histology xenografts by qRT-PCR (Mann Whitney test *p* = 0.0062). (**b**) Telomerase activity was significantly higher in xenografts with diffuse anaplasia when compared to favorable histology (*p* = 0.0021). (**c**) The relationship between percent blastema and telomerase activity is shown (linear regression R^2^ = 0.226, *p* = 0.001).

**Figure 4 cancers-14-01655-f004:**
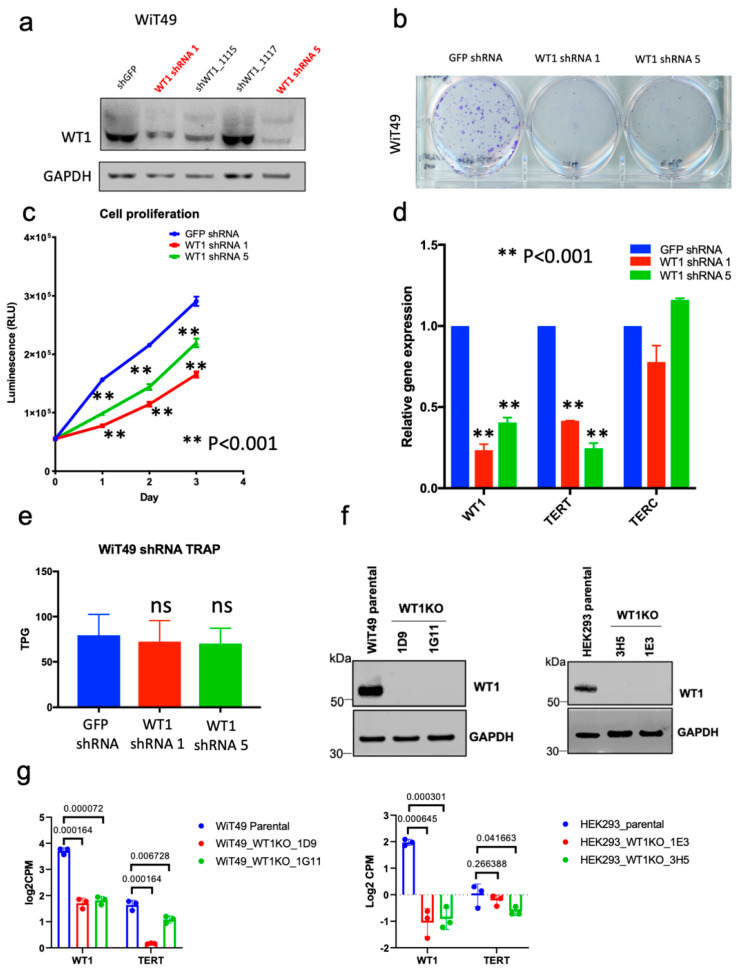
Functional association of WT1 and TERT in Wilms tumor cells in vitro. (**a**) shRNAs were knock down of *WT1* in anaplastic WiT49 cells. ShRNAs 1114 (shRNA 1) and 0596 (shRNA 5) were had adequate knockdown by western blot when compared to GFP non-targeting control. ShRNA knockdown of *WT1* was associated with (**b**) decreased colonies in a crystal violet assay and (**c**) cell proliferation at 1, 2, and 3 days (two-tailed *t*-test *p* < 0.001 all time points). (**d**) shRNA knockdown of *WT1* caused significant decrease in *TERT* expression (*p* < 0.001), but a nonsignificant reduction in (**e**) telomerase activity (TRAP assay). (**f**) Abrogation of WT1 protein detection in WiT49 and HEK293 *WT1* knockout clones. (**g**) CRISPR-Cas9-mediated knockout of *WT1* in WiT49 cells (WiT49-1D9 and WiT49-1G11 clones) and HEK293 cells (HEK293-1E3 and HEK293-3H5 clones) was associated with decreased *WT1* and *TERT* (*p* values shown for each clone).

**Figure 5 cancers-14-01655-f005:**
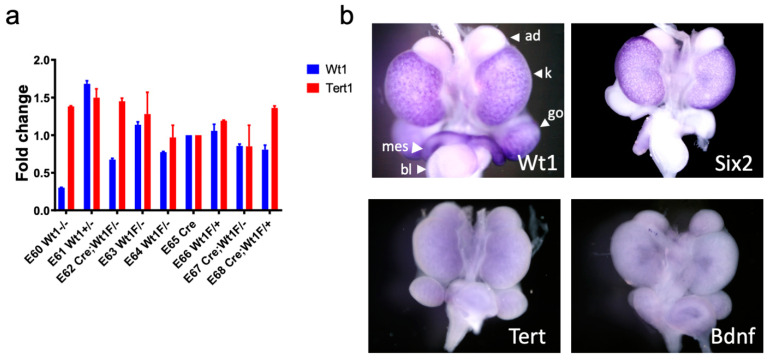
Tert and Wt1 are not related in kidney development. (**a**) RNA was isolated from murine embryonic kidneys and *Wt1* and *Tert* expression levels were compared by qRT-PCR. No relationship was seen between *Wt1* and *Tert* expression levels; most notably E60, E62, and E67 with biallelic loss of Wt1 did not exhibit reduction in *Tert* levels. (**b**) Whole mount in situ hybridization data from the GUDMAP database [34,35] demonstrates detection of *Wt1* (4× magnification) in the peripheral nephrogenic zone of e15 mouse embryonic kidneys (k) and is also seen in the gonads (go) and mesonephric ducts (mes). As a positive control, *Six2* (3.5× magnification) is also seen in the peripheral nephrogenic zone and in a distribution similar to *Wt1* in the e15 mouse embryonic kidneys. *Tert* (3.5× magnification) is not significantly detected in e15 kidneys. *Bdnf* (brain-derived neurotropic factor; 3.5× magnification) is shown as a negative control. Bladder-bl, Adrenal gland-ad.

**Figure 6 cancers-14-01655-f006:**
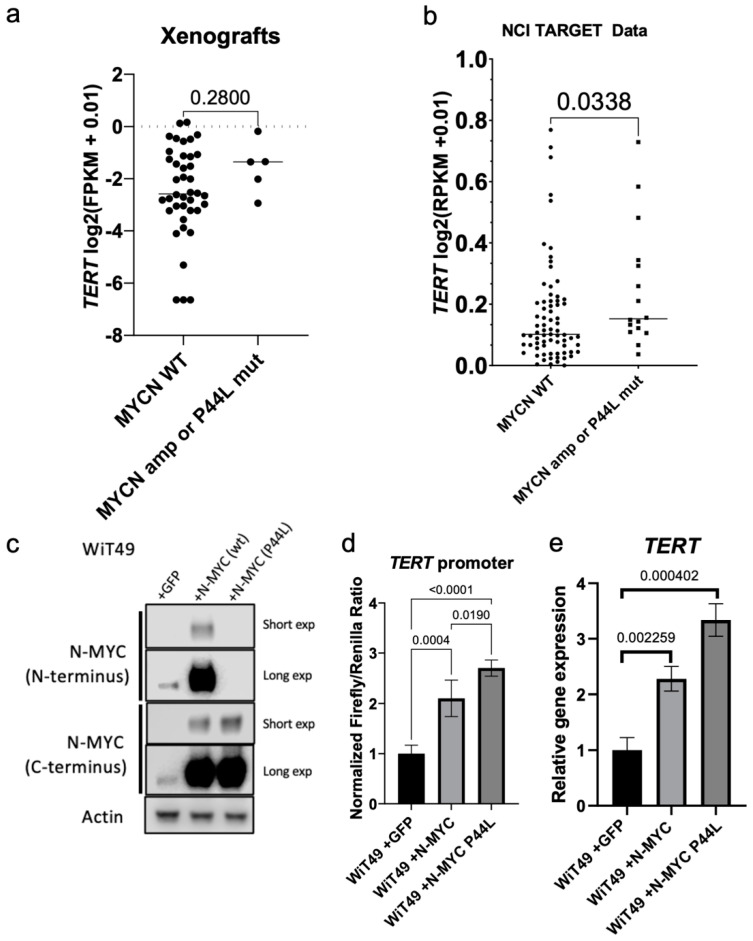
Relationship between *WT1* and *MYCN* in Wilms tumor. (**a**) *TERT* expression in *MYCN* wildtype xenografts versus *MYCN* amplified or P44L mutant samples (Mann Whitney test *p* = 0.280). (**b**) *TERT* expression in diploid, *MYCN* wildtype samples and those with *MYCN* amplification or P44L mutation in the NCI TARGET data (*p*= 0.0338). (**c**) Overexpression of wild type N-MYC and N-MYC P44L in WiT49 cells. The P44L mutation alters the recognized N-MYC (N-terminus) antibody epitope. The N-MYC (C-terminus) antibody detects overexpression of N-MYC in both conditions. Short exp—short exposure; Long exp—long exposure. Overexpression of wild-type N-MYC results in increased (**d**) *TERT* promoter activity and (**e**) *TERT* transcription by qRT-PCR. Overexpression of P44L mutant N-MYC results in increased (**d**) *TERT* promoter activity and (**e**) *TERT* transcription relative to wild type N-MYC and empty vector control (two-tailed *t*-test *p* values shown). Original labeled images of all Western blot and PCR gel data can be found at Appendix A.

**Figure 7 cancers-14-01655-f007:**
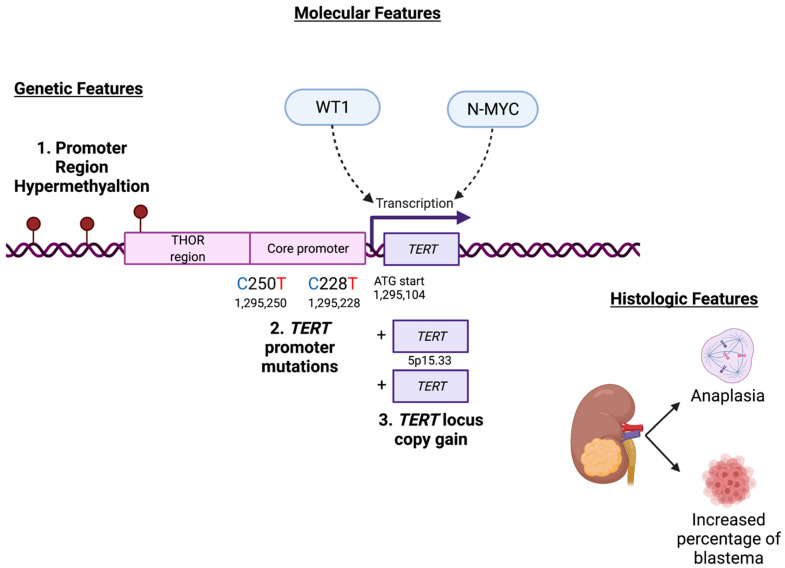
Summary of genetic, molecular, and histologic findings associated with *TERT* in Wilms tumor. Graphic created with BioRender.com.

**Table 1 cancers-14-01655-t001:** Characterization of CRISPR-Cas9 mediated WT1 knockout clones by whole exome sequencing. DEL—deletion mutation; INS—insertion mutation.

CRISPR-Cas9 WT1 KO Clone	Genomic Starting Coordinate GRCh37/hg19	Total Count	Alteration	Reference	Alteration	DEL	INS
WiT49_1D9	chr11:32,417,918	119 344	DEL INS	CAAGA CAAGA	C-AGA C**a**AAGA	A	A
WiT49_1G11	chr11:32,417,918	258	INS	CAAGA	C**a**AAGA		A
HEK293_3H5	chr11:32,417,918	117 226	DEL INS	CAAGA CAAGA	C-AGA C**a**AAGA	A	A
HEK293_1E3	chr11:32,417,918 chr11:32,417,919	93 130	INS DEL	CAAGA CAAGA	C**a**AAGA CA--A	AG	A

## Data Availability

The RNA-seq data from Wilms tumor patient derived xenografts are available in the European Genome-phenome Archive (EGA) database (https://www.ebi.ac.uk/ega/home) (accessed on 4 January 2022) under accession number EGAS00001003361. The Methylation EPIC beadchip array data used in this study are available in the Gene Expression Omnibus (GEO) database (https://www.ncbi.nlm.nih.gov/geo/) (accessed on 4 January 2022) under identifier GSE110697. All WTPDX used in this study are available to the scientific community upon request. Some results published here are based upon data generated by the Therapeutically Applicable Research to Generate Effective Treatments (https://ocg.cancer.gov/programs/target) (accessed on 4 January 2022) initiative, phs000218. The data used for this analysis are available at https://portal.gdc.cancer.gov/projects (accessed on 4 January 2022). We used models and data derived by the Human Cancer Models Initiative (HCMI) https://ocg.cancer.gov/programs/HCMI (accessed on 4 January 2022); dbGaP accession number phs001486. RNA-seq data comparing *TERT* expression across a multitude of pediatric cancers were obtained from St. Jude Cloud (https://www.stjude.cloud) (accessed on 4 January 2022) [23]. Whole mount in situ hybridization data were obtained from the GUDMAP database and are available at https://www.gudmap.org/ (accessed on 4 January 2022).

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
