# Peer review of "TERT Expression in Wilms Tumor Is Regulated by Promoter Mutation or Hypermethylation, WT1, and N-MYC"

_cancers, 2022, doi:10.3390/cancers14071655_

Round 1

Reviewer 1 Report

The authors started out from earlier work showing a higher risk status for Wilms tumors with elevanted TERT expression. They found evidence for TERT activation by promoter mutation, methylation or amplification in a set of xenografts. They then tried to nail down regulatory interaction based on loss of WT1 and MYCN gain or mutation as additional Wilms tumor drivers. To this end they provide evidence of WT1 driving TERT expression, perhaps through MYCN.

The manuscript is difficult to read. In several places it looks like a collection of experiments done in the wider area of a project on TERT, WT1 and Wilms tumor, but not like a stringent and coherent story to be told. Inconsistencies, questionable interpretations and lack of stringency lead to an impression of a papers that may not fully meet current publication standards.

Specific points of criticism are listed as they appear in the manuscript:

Line100: TERT expression in WT is claimed to be more modest compared to neuroblastoma and medulloblastoma. However, in the figure the latter appear to have a bimodal distribution with either lower or higher expression.

Line 162: It would be more informative to also compare primary tumors and xenografts to assess the stability of the methylation marks and to establish that results from both will be comparable in this respect.

The entire section 2.2 is overloaded with more or less redundant data and the value of Fig. 2 is limited. It is not helpful to bombard the reader with parallel analyses of expression by RNA-seq plus qRT-PCR. Furthermore, several of the WT1 wild-type samples show no or very low WT1 expression. It remains unclear whether they also have a concomitant reduction in TERT mRNA. The key message of Fig. 2 could be represented by a much more informative x-y plot of all WT1-TERT RNA-seq data, with highlighting of the WT1 mutant cases. Comparison of WT1 RNA with telomerase activity is less helpful to evaluate WT1 as a regulating transcription factor since it may involve additional layers of regulation (stability, TERC levels, …).

What is disturbing about Fig. 2G is the really low telomerase activity of the TERT mutant cases (10, 130, 180 units within the general span of 0-370), whereas these cases had significantly higher RNA expression (Fig. 1D). This must be commented on.

Fig 2H cannot be interpreted from the information given in the legend and may well be deleted. The weak literature mining (association) data in Fig. 2F should likewise be left out.

In a similar fashion Fig. 3 is overloaded. One of the panels A-C will suffice, and D-F are again highly redundant and D/E are mostly contained in Suppl. Fig. 4.

Line 199-201 is misleading since the tumors are presumably derived from treatment-naïve cases (COG protocol), where blastema is not a high-risk entity.

It is unclear why the experiment in Fig. 4F was performed (line 241-244). It is unrelated to Wilms tumors and does not add to the manuscript in any way.

Line 245: what is “the functional domain of WT1”? A much more precise description is needed as transcriptional activation and repression domains are also functional domains.

Fig. 4G: log2CPM is not explained and furthermore, it is strange to see “expression” of a gene inactivated by CRISPR/Cas9. Two steps on a log2-scale would only be a 4-fold difference in expression. Why did the authors switch from NB4 back to HEK293 cells that were not followed up before?

In the entire chapter 2.4 the knockout cell lines should be characterized first and then evaluated. The writing does not follow a logical path, but the sequence of data and experiments looks rather arbitrary, which is highly confusing.

The alterations induced by Cas9 in the different cell lines should be clearly described: where was the presumed cut, which indels followed and what would be the presumed effect on the transcript (frameshift, stop, NMD?).

The conclusion of chapter 2.4 cannot be drawn as stated in the text. The experiments were done in a WT-derived cell line with results that are difficult to explain (why should a zinc finger transcription factor without its DNA binding domain promote transcription of TERT? Not impossible but in need of additional explanations). Leaving the partial WT1 inactivation in WiT-49 and the unaffected HEK293 cells out, the data are based on a single cell clone and cannot be generalized to all Wilms tumors (with and without biallelic WT1 inactivation). Furthermore, these experiments have nothing to do with histological presentation (tumors without WT1 mutation may still resemble those carrying a mutation).

Chapter 2.5: The scope of the experiments is rather limited. As WT1 is differentially expressed in the various cell types in the developing kidney, the experiment may well lead to a regionally limited effect on TERT. But this can only be seen by in situ hybridization or other sensitive methods. Bulk kidney RNA can easily be non-informative in this regard. Furthermore, a lack of expression in scRNA-seq may not be relevant for genes expressed at low levels due to the much lower sequencing depth per cell.

The associated Fig. 5 is almost completely unnecessary except for part C (D-F just shows screenshots of a publicly available database with not enough relevant information; furthermore, it would have been appropriate to use available mouse data sets to explain a murine phenotype). The conclusions drawn by authors may be correct but are certainly not substantiated by the data presented.

Section 2.6/Fig. 6: There is no stringent case for WT1 regulating MYCN transcriptionally. The even stronger down-regulation of MYCN in clone 1G11 (Fig 6D) contrasts sharply with the opposite picture in Fig. 6E, where MYCN protein is at best non-regulated, if not induced. The argument that this clone may express a truncated version cannot be employed to just support any deviation from what was expected and internal inconsistencies.

In Fig. 6F the unexpected interaction of WT1 and MYCN proteins should be supported by reciprocal IPs. Furthermore, extensive MS-based MYC interactome studies with hundreds of partner proteins, sometimes using the same HEK293 cells failed to identify WT1 as a binder, which is very surprising given the strong interaction shown by the authors. This should be explained thoroughly.

The following Figure 6G is further confusing since a strong overexpression of MYCN-wt is seen with a C-terminal antibody, while the N-Terminal antibody suggests equal loading. This control makes the entire experiment useless. The complete lack of staining of the P44L mutant with one of the antibodies may be due to the epitope encompassing the mutant position, but there is no mention at all of this obvious discrepancy.

The discussion is quite long and repeats the results section with numerous speculations that are not supported by the data presented. Correlations are often interpreted as causality.

Two global, but minors issues

The MYCN gene encodes the protein N-MYC, even if this is not intuitive, it is current nomenclature. Thus, the gene name MYCN should be used throughout instead of NMYC.

Similarly, HEK293 cells are derived from human embryonic kidneys, but they are of neural origin, which should always be remembered. In the present manuscript, where readers may be misled into believing that these cells are kidney precursors, this should explicitly be stated.

Author Response

Reviewer 1 Comments:

The authors started out from earlier work showing a higher risk status for Wilms tumors with elevanted TERT expression. They found evidence for TERT activation by promoter mutation, methylation or amplification in a set of xenografts. They then tried to nail down regulatory interaction based on loss of WT1 and MYCN gain or mutation as additional Wilms tumor drivers. To this end they provide evidence of WT1 driving TERT expression, perhaps through MYCN.

The manuscript is difficult to read. In several places it looks like a collection of experiments done in the wider area of a project on TERT, WT1 and Wilms tumor, but not like a stringent and coherent story to be told. Inconsistencies, questionable interpretations and lack of stringency lead to an impression of a papers that may not fully meet current publication standards.

Response: Thank you for your thorough and critical review of our manuscript. An itemized list of responses to each of your comments is included below.

Specific points of criticism are listed as they appear in the manuscript:

Line100: TERT expression in WT is claimed to be more modest compared to neuroblastoma and medulloblastoma. However, in the figure the latter appear to have a bimodal distribution with either lower or higher expression.

Response: Medulloblastoma and neuroblastoma are known to have a bimodal distribution of TERT expression according to their molecular subtype. This has been added to the manuscript: “Median expression of TERT in Wilms tumor was found to be between the two modes of TERT expression seen in neuroblastoma and medulloblastoma, which have bimodal TERT expression according to the tumor molecular subtype (Fig. 1b)[18, 38]. Overall, these data show that a small proportion of Wilms tumor samples have high TERT expression.”

Line 162: It would be more informative to also compare primary tumors and xenografts to assess the stability of the methylation marks and to establish that results from both will be comparable in this respect.

Response: Violin plots depicting the distribution of methylation patterns at each of the three TERT-promoter related sites in primary tumors (PT), xenografts (KT), and normal kidney (NK) have now been included in Figure 1 to enable better comparison of the stability of methylation marks between primary tumors and xenografts. Although statistically significant increases are depicted when xenografts are compared to primary tumors, the overall pattern of methylation is preserved and remains notably distinct from the much lower levels found in normal kidney specimens.

The entire section 2.2 is overloaded with more or less redundant data and the value of Fig. 2 is limited. It is not helpful to bombard the reader with parallel analyses of expression by RNA-seq plus qRT-PCR. Furthermore, several of the WT1 wild-type samples show no or very low WT1 expression. It remains unclear whether they also have a concomitant reduction in TERT mRNA. The key message of Fig. 2 could be represented by a much more informative x-y plot of all WT1-TERT RNA-seq data, with highlighting of the WT1 mutant cases. Comparison of WT1 RNA with telomerase activity is less helpful to evaluate WT1 as a regulating transcription factor since it may involve additional layers of regulation (stability, TERC levels, …).

Response: RNA-seq data have been moved to the supplemental figures to eliminate redundancy in Figure 2. The entire figure has now been consolidated to 4 panels considering the reviewer’s comments. The suggested X-Y plot showing WT1 and TERT expression data has now replaced the WT1 expression-Telomerase activity X-Y plot from the original version with WT1-mutant, TERT promoter mutant, and anaplastic samples noted on the graphic.

What is disturbing about Fig. 2G is the really low telomerase activity of the TERT mutant cases (10, 130, 180 units within the general span of 0-370), whereas these cases had significantly higher RNA expression (Fig. 1D). This must be commented on.

Response: The xenografts with TERT promoter mutations have been specifically indicated in the revised version of Figure 2 (Figure 2D) so that the reader can see the relative expression of TERT in these xenografts when compared to the remainder of samples. We have added the following sentence to results section 2.1: “However, in our xenograft data set, the three TERT promoter mutant specimens were not found to have significantly higher TERT expression or telomerase activity than non-mutant specimens.” Due to the limited sample size (n=3), we cannot determine without speculating why there seems to be a discrepancy between TERT expression and telomerase activity in these three specimens; however, in our entire dataset TERT expression and telomerase activity were found to be highly correlated (r=0.72, p=3.3 x 10-8) as shown in Supplementary Figure 4.

Fig 2H cannot be interpreted from the information given in the legend and may well be deleted. The weak literature mining (association) data in Fig. 2F should likewise be left out.

Response: Panels 2H and 2F have been eliminated from Figure 2.

In a similar fashion Fig. 3 is overloaded. One of the panels A-C will suffice, and D-F are again highly redundant and D/E are mostly contained in Suppl. Fig. 4.

Response: We have limited Figure 3 to 3 panels (TERT q-RT-PCR, telomerase activity, percent blastema analysis) that are no longer redundant. We feel that showing telomerase activity in addition to TERT expression are orthogonal assays to assess the same pathway and are not redundant.

Line 199-201 is misleading since the tumors are presumably derived from treatment-naïve cases (COG protocol), where blastema is not a high-risk entity.

Response: The comment about blastemal predominance has been eliminated from the title of this section. The paragraph has been edited to eliminate the term blastemal predominance, and rather focus on the correlation between the percent blastema and the telomerase activity. The term blastemal predominance has been eliminated from Figure 7 (Summary Figure).

It is unclear why the experiment in Fig. 4F was performed (line 241-244). It is unrelated to Wilms tumors and does not add to the manuscript in any way.

Response: We performed the experiment in Figure 4F because the NB4 cell line is one of the only other malignant cancer lines available that expresses high levels of WT1. We wanted to confirm that knockdown of WT1 resulted in decreased expression of TERT in another malignant cell line since only one of our Wilms tumor cell lines expressed WT1. However, we agree that it is not core to the message of the manuscript and thus it has been eliminated from Figure 4F and moved to the supplemental data.

Line 245: what is “the functional domain of WT1”? A much more precise description is needed as transcriptional activation and repression domains are also functional domains.

Response: The description has been rewritten to include more precise language: “Using CRISPR-Cas9-mediated disruption of the second and third of four DNA-binding zinc-finger domains of WT1, we created WT1-null WiT49 cells…”

Fig. 4G: log2CPM is not explained and furthermore, it is strange to see “expression” of a gene inactivated by CRISPR/Cas9. Two steps on a log2-scale would only be a 4-fold difference in expression. Why did the authors switch from NB4 back to HEK293 cells that were not followed up before?

In the entire chapter 2.4 the knockout cell lines should be characterized first and then evaluated. The writing does not follow a logical path, but the sequence of data and experiments looks rather arbitrary, which is highly confusing.

Response: Log2CPM is the 2-base logarithmic transformation of counts per million reads detected for a given gene. In the St. Jude Cloud WARDEN workflow used for the in vitro cell line analysis, log2CPM is the output used to quantify gene expression. This information has been added to the manuscript in the methods section: “Using the WARDEN workflow, log2 counts per million (log2CPM) values were used to quantify gene expression according to mapped RNA-seq reads.” The strategy we used for CRISPR-Cas9-mediated knockout of WT1 in this manuscript was to design guide RNAs targeting WT1 exon 7, which mirrors the strategy used to knockout Wt1 in murine models with well-documented phenotypes consistent with loss of Wt1 function. However, as we have shown in part, we believe that in the anaplastic WiT49 Wilms tumor cell line, N-terminal fragments remain expressed. RNA-seq based analyses is likely picking up reads from the N-terminal portion of WT1 as we have shown in the supplementary data. This may explain the less than expected reduction in transcript detection in the WiT49 cell line. Based on these findings, extensive characterization of the N-terminal portion of WT1 in the context of TP53 mutation will be the subject of future research by our group, but we believe that is beyond the scope of the current manuscript.

We thought that using HEK293 for WT1 knockout might be more relevant to our understanding of Wilms tumor biology because of Wilms tumor’s hypothesized developmental origin in the embryonic kidney rather than NB4, which is a hematogenous malignancy. However, the HEK293 cell line has the known limitations pointed out by the reviewer in the comment below that have been added to the limitations section of the manuscript.

The writing of this section has been revised to follow a more logical flow.

The alterations induced by Cas9 in the different cell lines should be clearly described: where was the presumed cut, which indels followed and what would be the presumed effect on the transcript (frameshift, stop, NMD?).

Cell lines have been characterized by whole exome sequencing. Both clones for HEK293 and both clones for WiT49 exhibit a frameshift mutation caused by insertion and/or deletion in exon 7 of WT1. This information has been added to the manuscript in Table 1.

The conclusion of chapter 2.4 cannot be drawn as stated in the text. The experiments were done in a WT-derived cell line with results that are difficult to explain (why should a zinc finger transcription factor without its DNA binding domain promote transcription of TERT? Not impossible but in need of additional explanations). Leaving the partial WT1 inactivation in WiT-49 and the unaffected HEK293 cells out, the data are based on a single cell clone and cannot be generalized to all Wilms tumors (with and without biallelic WT1 inactivation). Furthermore, these experiments have nothing to do with histological presentation (tumors without WT1 mutation may still resemble those carrying a mutation).

Response: Our goal in this manuscript is to only make conclusion statements that are supported by the data and to clearly state the limitations of our approaches. Therefore, at the reviewer’s suggestion, this conclusion statement has been eliminated from the manuscript.

Chapter 2.5: The scope of the experiments is rather limited. As WT1 is differentially expressed in the various cell types in the developing kidney, the experiment may well lead to a regionally limited effect on TERT. But this can only be seen by in situ hybridization or other sensitive methods. Bulk kidney RNA can easily be non-informative in this regard. Furthermore, a lack of expression in scRNA-seq may not be relevant for genes expressed at low levels due to the much lower sequencing depth per cell.

The associated Fig. 5 is almost completely unnecessary except for part C (D-F just shows screenshots of a publicly available database with not enough relevant information; furthermore, it would have been appropriate to use available mouse data sets to explain a murine phenotype). The conclusions drawn by authors may be correct but are certainly not substantiated by the data presented.

Response: Figures 5A and 5B have been moved to the supplementary data. At the reviewer’s suggestion, publicly available data from mouse data sets have been queried to expand our understanding of the observed murine phenotype. Using publicly available data from the GUDMAP database, in situ hybridization data for Wt1 and Tert have been added to the figure. These data support our findings that Tert and Wt1 are not related in murine kidney development. The human fetal kidney data have been eliminated at the reviewer’s suggestion.

Section 2.6/Fig. 6: There is no stringent case for WT1 regulating MYCN transcriptionally. The even stronger down-regulation of MYCN in clone 1G11 (Fig 6D) contrasts sharply with the opposite picture in Fig. 6E, where MYCN protein is at best non-regulated, if not induced. The argument that this clone may express a truncated version cannot be employed to just support any deviation from what was expected and internal inconsistencies.

Response: We performed additional experiments to assess the role of WT1 regulating MYCN more transcriptionally. WT1 knockdown in HEK293 cells resulted in decreased transcription of MYCN and decreased protein expression of N-MYC. However, in WiT49 we did not observe this phenomenon. This could be due to WiT49 having MYCN copy number gain/amplification. In any case, we are unable to unequivocally support WT1 transcriptionally regulating MYCN in Wilms tumor and therefore we have removed data linking WT1 to the regulation of MYCN from the manuscript. We have instead focused only on the link between MYCN and TERT, as our data support the known role of MYCN in regulating TERT transcription.

In Fig. 6F the unexpected interaction of WT1 and MYCN proteins should be supported by reciprocal IPs. Furthermore, extensive MS-based MYC interactome studies with hundreds of partner proteins, sometimes using the same HEK293 cells failed to identify WT1 as a binder, which is very surprising given the strong interaction shown by the authors. This should be explained thoroughly.

Response: We attempted the recommended reciprocal IP using 4 different WT1 antibodies. We did not observe technically successful pull-down of WT1. Therefore, we transiently expressed FLAG-tagged WT1 isoforms A-D and did an IP for FLAG with western blot for NMYC and WT1. We did not observe protein-protein interaction between WT1 and N-MYC using this method. Therefore, we are unable to corroborate our IP results using an independent assay. We have therefore removed the data regarding protein-protein interaction between WT1 and N-MYC from the manuscript.

The following Figure 6G is further confusing since a strong overexpression of MYCN-wt is seen with a C-terminal antibody, while the N-Terminal antibody suggests equal loading. This control makes the entire experiment useless. The complete lack of staining of the P44L mutant with one of the antibodies may be due to the epitope encompassing the mutant position, but there is no mention at all of this obvious discrepancy.

Response: The P44L mutation disrupts the epitope recognized by the anti-N-MYC Santa Cruz (sc) antibody used in this study. In contrast, the Cell Signaling Technologies anti-N-MYC (D1V2A) antibody is able to recognize both wild-type N-MYC and N-MYC p44L. This has been clarified in the manuscript. The western blot has been repeated with clearer results displayed.

The discussion is quite long and repeats the results section with numerous speculations that are not supported by the data presented. Correlations are often interpreted as causality.

Response: The discussion has been revised with these comments in mind. The discussion section has been shortened. An effort has been made to reduce speculation in the discussion section.

Two global, but minors issues

The MYCN gene encodes the protein N-MYC, even if this is not intuitive, it is current nomenclature. Thus, the gene name MYCN should be used throughout instead of NMYC.

Response: NMYC has been changed to MYCN throughout the manuscript when referring to the gene. N-MYC is now used when referring to the protein throughout the manuscript.

Similarly, HEK293 cells are derived from human embryonic kidneys, but they are of neural origin, which should always be remembered. In the present manuscript, where readers may be misled into believing that these cells are kidney precursors, this should explicitly be stated.

Response: This has now been clearly noted in the limitations paragraph of the discussion section with an appropriate reference provided.

Reviewer 2 Report

The authros submit a very elaborate, comprehensive study on Wilms tumor biology, focussing on the biological network between TERT expression/telomerase activity, and the oncogenes MYCN and WT1. The authors add significant new knowledge to thescientific discussion, and present their experimental data in a clear way. Of not, the authors are able to correlate their experimental findings with clinical and histological patterns of prognostic importance, e.g. anaplastic histology, blastemal differentiation. The experimental approach is sound.

From a clinical perspective, I would ask the authros, whether they could provide a list of the clinical and pathological features of the tumor samples studied. it would be interesting to the clinical audience, whether tumors with germ line WT1 mutation or tumors arising in the context of Beckwith-Wiedemann syndrome have also been studied. In this context I would appreciate if they authors could also allude to the two main different biological groups of WT, e.g. WT1 mutation associated WT and imprinting error (e.g. BWS) associated WT. 

Overall, great study!

Author Response

Reviewer 2 Comments:

The authros submit a very elaborate, comprehensive study on Wilms tumor biology, focussing on the biological network between TERT expression/telomerase activity, and the oncogenes MYCN and WT1. The authors add significant new knowledge to thescientific discussion, and present their experimental data in a clear way. Of not, the authors are able to correlate their experimental findings with clinical and histological patterns of prognostic importance, e.g. anaplastic histology, blastemal differentiation. The experimental approach is sound.

From a clinical perspective, I would ask the authros, whether they could provide a list of the clinical and pathological features of the tumor samples studied. it would be interesting to the clinical audience, whether tumors with germ line WT1 mutation or tumors arising in the context of Beckwith-Wiedemann syndrome have also been studied. In this context I would appreciate if they authors could also allude to the two main different biological groups of WT, e.g. WT1 mutation associated WT and imprinting error (e.g. BWS) associated WT. 

Response: Supplementary Figure 1 shows a heatmap summary of clinical data including gender, disease stage, histology, disease relapse, and death. It also shows status of 1q gain, 1p LOH, 16q LOH, WT1 mutation, NMYC copy number/mutation, TERT promoter mutation, TERT promoter methylation status, and TERT copy number. At the reviewer’s suggestion, we have added 11p15 status to this heatmap to include WT associated with 11p15 imprinting errors. No patient in the current cohort had Beckwith Wiedemann syndrome. Supplementary Figure 3 contains detailed information about the WT1 mutation and expression status of each xenograft model.

Overall, great study!

Response: Thank you for your review of our manuscript.

Reviewer 3 Report

Authors investigate regulation of TERT expression in Wilms tumor. Regulation of TERT expression and telomerase activity by WT1 and NMYC is studied using in vitro and in vivo ( inducible cKO ) models. TERT expression and correlation with outcome in solid tumor is very relevant topic since telomerase targeted therapies could be used in these patients. Only small number of WT patients have high TERT expression. It is still relevant and this study is very well conducted and presented in this manuscript. This study adds to the growing body of literature around TERT expression, TERT expression regulation and TERT targeting in various cancers. The figures as presented now are not clearly legible. Consider making figure labels more legible, larger font size.

Author Response

Reviewer 3 Comments:

Authors investigate regulation of TERT expression in Wilms tumor. Regulation of TERT expression and telomerase activity by WT1 and NMYC is studied using in vitro and in vivo ( inducible cKO ) models. TERT expression and correlation with outcome in solid tumor is very relevant topic since telomerase targeted therapies could be used in these patients. Only small number of WT patients have high TERT expression. It is still relevant and this study is very well conducted and presented in this manuscript. This study adds to the growing body of literature around TERT expression, TERT expression regulation and TERT targeting in various cancers. The figures as presented now are not clearly legible. Consider making figure labels more legible, larger font size.

Response: We agree that a small proportion of Wilms tumor patients have high expression of TERT. This is one reason we wanted to show Figure 1 panel B to inform the reader of the expression of TERT in Wilms tumor relative to other pediatric cancers. In consideration of the reviewers comments, we have added the following sentences to section 2.1 of the results section: “Median expression of TERT in Wilms tumor was found to be between the two modes of TERT expression seen in neuroblastoma and medulloblastoma, which have bimodal TERT expression according to the tumor molecular subtype (Fig. 1B)[18, 21]. Overall, these data show that a small proportion of Wilms tumor samples have high TERT expression.”

Response: We have eliminated redundant information in the figure panels in order to make the remaining panels bigger in several figures throughout the manuscript.

Reviewer 4 Report

Following are a few suggestions to the authors to revise the manuscript.

  1. Figure should be labeled as Fig. 1 and all the figures and tables should be cited in the text in the manuscript. Kindly recheck.
  2. All the figures should be labeled if it has subgroups in small letter (a), (b), (c) clearly appearing on the figures and provide the high-resolution image files with the clarity of the text in the figure with better readability.
  3. Avoid writing we, our study, our findings in the paper instead use as "the present study"
  4.  The discussion section in the manuscript lacks the comparison of the previous relevant similar studies and their results. The authors should cite the recent similar studies and compare the efficacy with the present proposed methodology and the critical outcomes. 
  5. The novelty of the study should be mentioned in the abstract as well as in the ending note of the introduction.
  6.  The main sections should be Introduction, Methodology or Materials and Methods, Results and Discussion, Conclusion
  7. The short form should be avoided in the title.
  8. Website link in the main manuscript hyperlink can be removed as citation of the website is included in the references
  9. The reactions and DNA Sequencing can be represented as an equation for better visibility and understanding for the readers.

Author Response

Reviewer 4 Comments:

Following are a few suggestions to the authors to revise the manuscript.

  1. Figure should be labeled as Fig. 1 and all the figures and tables should be cited in the text in the manuscript. Kindly recheck.

Response: We have made this change and used this abbreviation throughout the manuscript. We have ensured that all figures and tables are cited appropriately throughout the manuscript. 

  1. All the figures should be labeled if it has subgroups in small letter (a), (b), (c) clearly appearing on the figures and provide the high-resolution image files with the clarity of the text in the figure with better readability.

Response: The figures have been modified accordingly and high-resolution versions of the figures have been uploaded to accompany the revised manuscript.

  1. Avoid writing we, our study, our findings in the paper instead use as "the present study"

Response: The use of “we” and “our” has been eliminated throughout the manuscript.

  1.  The discussion section in the manuscript lacks the comparison of the previous relevant similar studies and their results. The authors should cite the recent similar studies and compare the efficacy with the present proposed methodology and the critical outcomes. 

Response: The discussion section has been revised with this and the above reviewer’s comments in consideration.

  1. The novelty of the study should be mentioned in the abstract as well as in the ending note of the introduction.

Response: A comment on the novelty of the study has been added to both the abstract and the ending of the introduction. This study is novel because it identifies several mechanisms of TERT activation in Wilms tumor that could be of therapeutic interest in the future.

  1.  The main sections should be Introduction, Methodology or Materials and Methods, Results and Discussion, Conclusion

Response: The sections of the manuscript have been re-ordered accordingly.

  1. The short form should be avoided in the title.

Response: Other than commonly used gene names that facilitate search strategies for this manuscript, no abbreviations are used in the title.

  1. Website link in the main manuscript hyperlink can be removed as citation of the website is included in the references

Response: Hyperlinks have been removed from the main body of the manuscript. There are a couple hyperlinks in the data availability statement which we believe helps interested readers obtain available data.

  1. The reactions and DNA Sequencing can be represented as an equation for better visibility and understanding for the readers.

Response: Custom primer sequences have now been formatted as equations to improve visibility and understanding for the readers.